# Bayes-inspired Integration of Pretrained Priors and Few-Shot Evidence for Few-Shot Classification

**Mingyang Zhou** [1]  **Xiaoxuan Zhang** [1]  **Gang Liu** [* 1]  **Yuhong Feng** [1]  **Xiaoqun Wu** [1]  **Hao Liao** [* 1]  **Rui Mao** [1]

## Abstract

Few-shot classification aims to adapt a pretrained model to novel classes with limited examples. While current methods often heuristically combine pretrained knowledge and few-shot evidence, we seek a more principled understanding of their relationship. In this paper, we propose a Bayesian-inspired optimal integration framework(BOIF) that interprets pretrained models as priors and few-shot evidence as likelihoods. Under conditional independence approximation, we show that the optimal log-posterior decomposes into the sum of prior logits and likelihood logits. This leads to a simple yet effective design principle: decouple the prior and likelihood pathways and combine their logits additively. Guided by this principle, we implement BOIF using CLIP with two novel enhancements: (1) a multi-level feature adapter to enrich visual representations, and (2) a simplified cache module for likelihood estimation. Extensive experiments on 11 benchmarks show BOIF achieves state-of-the-art performance (e.g., 80.61% average accuracy at 16-shot) and strong out-of-distribution robustness. Our work provides both a principled perspective and an effective instantiation for few-shot adaptation.

## 1. Introduction

Deep learning has achieved remarkable success in recent years, fueled largely by the availability of massive labeled datasets. However, in many real-world scenarios—such as medical diagnosis, rare species identification, or personalized recommendation—acquiring sufficient labeled data is often prohibitively expensive or practically impossible (Ali et al., 2026). This challenge has catalyzed the development of Few-Shot Learning (FSL), which aims to empower models to recognize novel categories given only a limited number of support examples. Traditional FSL approaches, such as Model-Agnostic Meta-Learning (MAML) (Finn et al., 2017) and Prototypical Networks (Snell et al., 2017), primarily follow a "learning to learn" paradigm, simulating few-shot tasks during training to acquire transferable optimization strategies or metric spaces (Song et al., 2026). However, the advent of large-scale foundation models has precipitated a significant paradigm shift. Vision-Language Models (VLMs) like CLIP (Radford et al., 2021), pre-trained on billion-scale image-text pairs, possess robust open-world semantic priors. Consequently, the prevailing practice in FSL has evolved from training meta-learners from scratch to a "pre-train, then adapt" paradigm, which focuses on effectively adapting these powerful frozen priors to downstream tasks (Tsoumplekas et al., 2025).

To adapt pre-trained models to new downstream tasks, various Parameter-Efficient Fine-Tuning (PEFT) and training-free methods have been proposed, including Prompt Tuning (Zhou et al., 2022b;a) and Adapter-based methods (Zhang et al., 2022; Gao et al., 2024). Nevertheless, to the best of our knowledge, the integration of pre-trained knowledge (prior) and few-shot evidence (support set) remains predominantly heuristic. Most existing methods fuse these two information sources through ad-hoc architectural designs—such as residual connections or simple weighted summation of outputs—relying heavily on empirical trial-and-error to determine optimal configurations. Consequently, current adaptation network designs reflect engineering intuition rather than rigorous theoretical derivation. This limitation leads us to our primary research question:

**Question 1**: *What is the relationship between pre-trained knowledge and few-shot evidence, and how can we integrate few-shot evidence into pre-trained models?*

In this paper, we bridge this gap by introducing a novel Bayesian perspective to the problem of few-shot adaptation. We formulate the pre-trained model as an approximation of the Bayesian posterior conditioned on the pre-training dataset, and the few-shot adaptation process as the estimation of the posterior conditioned on the downstream few-shot set. Within this framework, we theoretically demon-

---

[1]College of Computer Science and Software Engineering, Shenzhen University, Shenzhen, P. R. China. Correspondence to: Gang Liu <gliu@szu.edu.cn>, Hao Liao <haoliao@szu.edu.cn>.

*Proceedings of the 43$^{rd}$ International Conference on Machine Learning*, Seoul, South Korea. PMLR 306, 2026. Copyright 2026 by the author(s).

strate that the posterior logits are derived from the summation of the output logits of the pre-trained model and those of the few-shot evidence. Specifically, the pre-trained knowledge and the few-shot evidence are treated as independent components that contribute equally to the posterior logits. Guided by this theoretical insight, we propose the Bayesian-inspired Optimal Integration Framework (BOIF), which decomposes the classification problem into two decoupled pathways: (1) a *Pre-trained Prior Module* that leverages the pre-trained model's zero-shot capability to generate pre-trained posterior logits; and (2) a *Support-Set Likelihood Module* that only utilizes a few-shot-based classifier to generate few-shot evidence logits. Unlike previous heuristic approaches, BOIF designs them independently and integrates them via a theoretically grounded logit-summation rule. Our main contributions are:

- **A Bayesian design principle** for few-shot learning: we derive an additive logit combination rule from conditional independence assumptions, providing a principled alternative to heuristic fusion strategies.

- **A decoupled dual-path architecture** implementing this principle: unlike previous works that combines pretrained and support-set components via heuristic methods, BOIF explicitly separates them into a prior pathway and a likelihood pathway, then combines via parameter-free equal-weight logit addition.

- **State-of-the-art performance** across 11 datasets and strong out-of-distribution robustness, validated by extensive comparisons and ablation studies.

## 2. Related Work

**Pretrained Foundation Models.** The paradigm of computer vision and natural language has shifted from training task-specific models to leveraging pretrained foundation models. Early approaches relied on supervised pre-training over massive datasets like ImageNet (Deng et al., 2009), learning generic feature representations that transfer effectively to downstream tasks (He et al., 2016; Kornblith et al., 2019). Recently, self-supervised learning methods(e.g., MoCo (He et al., 2020), SimCLR (Chen et al., 2020), MAE (He et al., 2022), BEiT (Bao et al., 2022)) emerged to mitigate the dependency on annotated data by constructing robust feature spaces from unlabeled images. More recently, the field has witnessed a breakthrough with Vision-Language Models (VLMs) (Li et al., 2022; Yang et al., 2023b; Dai et al., 2024; Wang et al., 2024; OLMo et al., 2025) like CLIP (Radford et al., 2021) and ALIGN (Jia et al., 2021). By aligning visual features with open-world semantic concepts, these models achieve remarkable zero-shot prediction capabilities (Zhou et al., 2023; Zhang et al., 2025). However, downstream tasks often exhibit signifi-

cant distributional shifts compared to the pretraining data, leading to suboptimal performance (Zhu et al., 2025). Consequently, developing effective fine-tuning strategies has become a challenge in recent research.

**Few-Shot Learning.** Few-shot learning aims to enable models to recognize previously unseen categories given a limited number of examples. Early research primarily followed the meta-learning paradigm, including optimization-based methods (Gao et al., 2021; Ke et al., 2024; Fu et al., 2025) like MAML (Finn et al., 2017) and metric-based approaches (Musgrave et al., 2020; Jung et al., 2022; Lv et al., 2025) such as Prototypical Networks (Snell et al., 2017). With the advent of foundation models, the focus has transitioned towards the "pretrain-then-adapt" paradigm (Tsoumplekas et al., 2025). To adapt foundation models efficiently, Parameter-Efficient Fine-Tuning methods have been widely adopted, including LoRA (Zanella & Ben Ayed, 2024; Luo et al., 2025; de Vera et al., 2025; Ji et al., 2026), prompt learning (Lee et al., 2023a; Miyai et al., 2023; Miao et al., 2025; Wang et al., 2025), prefix tuning (Wang et al., 2023; Yuan et al., 2024), and adapter-based methods (Zhou & Zou, 2025; Ye et al., 2026). In particular, prompt learning techniques, such as CoOp (Zhou et al., 2022b), Co-CoOp (Zhou et al., 2022a), and their variants (Jia et al., 2022; Zhu et al., 2023), optimize learnable vectors to adapt the semantic space. Adapter-based methods inject lightweight modules into foundation models, such as residual transformations in CLIP-Adapter (Gao et al., 2024) or cache-based inference in Tip-Adapter (Zhang et al., 2022) and SuS-X (Udandarao et al., 2023). However, most existing methods integrate pretrained knowledge and few-shot information through heuristic designs (Zhang et al., 2022)–lacking a rigorous analysis of how to combine these two information sources.

**Feature Enhancement and Representation Learning.** While pretrained models offer robust zero-shot priors, relying solely on final-layer representations often yields sub-optimal performance due to the feature distribution shift in downstream data (Rao et al., 2022; Hu et al., 2022; Jia et al., 2025; Gopalbhai & N. Modi, 2025). To address this limitation, feature enhancement has emerged as a pivotal research direction. In computer vision, hierarchical architectures such as Feature Pyramid Networks (FPN) (Lin et al., 2017; Yang et al., 2022; 2023a) and U-Net (Ronneberger et al., 2015; Park et al., 2025) utilize lateral skip-connections to fuse high-resolution low-level features. Beyond architectural modifications, transfer learning approaches aggregate activations from intermediate layers to construct denser and more discriminative descriptors (Salih et al., 2025). Consequently, incorporating multi-level features significantly benefits accurate few-shot inference.

Different from existing heuristic few-shot learning methods,

we provide an analytical relationship between pretrained models and few-shot evidence within the classification task. Based on Bayesian theory, we prove that few-shot classification relies on the independent contributions of the two components, which is different from existing Bayesian-based methods that either focus on learning a better initial parameters distribution for new tasks (e.g., Bayesian MAML (Yoon et al., 2018)) or use Bayesian networks as the pretrained neural models (e.g., Bayesian prototypical networks (Gao et al., 2025)). We then introduce a feature enhancement framework to refine the representations of the two components, offering a deep understanding of designing effective few-shot models.

# 3. Methodology

## 3.1. Problem Statement

In few-shot classification, we consider a limited support set of labeled examples $\mathcal{D}_{train} = \{(\mathbf{x}_i, y_i)\}_{i=1}^{C \times K}$, where each $\mathbf{x}_i \in \mathbb{R}^D$ denotes a D-dimensional feature vector and $y_i \in \mathcal{Y}_{novel} = \{1, 2, ..., C\}$ is its corresponding class label. The training set contains $K$ samples per class, yielding a total of $C \times K$ examples. In the few-shot setting, $K$ is typically very small, making it infeasible to train an effective classification model solely from $\mathcal{D}_{train}$. Instead, a common practice is to fine-tune a pretrained model $f_\theta(\cdot)$ (parameterized by $\theta$) on $\mathcal{D}_{train}$ by minimizing the cross-entropy loss:

$$\mathcal{L}_{CE} = -\frac{1}{C \cdot K} \sum_{(\mathbf{x}_i, y_i) \in \mathcal{D}_{train}} \log P(y_i | \mathbf{x}_i; \theta), \quad (1)$$

where $P(y_i | \mathbf{x}_i; \theta)$ denotes the predicted probability assigned by the model to the true label $y_i$. To evaluate the model's performance, a test set $\mathcal{D}_{test} = \{(\mathbf{x}_j, y_j)\}_{j=1}^N$ is used, where $N$ is the total number of test samples. For each test sample $(\mathbf{x}_j, y_j)$, the fine-tuned model produces a prediction $\hat{y}_j = argmax_{y \in \{1, 2, ..., C\}} P(y | \mathbf{x}_j; \theta)$. The classification accuracy(ACC) over the test set is computed as $ACC = \frac{1}{N} \sum_{j=1:N} \mathbb{1}(\hat{y}_j = y_j)$, where $\mathbb{1}(\hat{y}_j = y_j)$ is the indicator function, equal to 1 if the prediction matches the ground truth and 0 otherwise. Note that $\mathcal{D}_{test}$ shares the same $C$ classes as $\mathcal{D}_{train}$.

**Challenges:** While several approaches exist for fine-tuning pretrained models in few-shot settings–including retraining from scratch, Prototypical Networks (Snell et al., 2017), and meta-learning (Finn et al., 2017)–most current methods rely heavily on heuristic design. How to design the fine-tuning framework is still an open question. This motivates our work to develop a general framework that formally analyzes the theoretical relationship between a pretrained model and the support set, and systematically optimizes their combination for improved few-shot learning.

## 3.2. A Bayesian Perspective on Few-shot Adaptation

**Motivation.** Existing few-shot methods combine pretrained models and support sets through heuristics like residual connections or weighted averaging. To better understand this combination, we formulate the problem from a Bayesian viewpoint. Consider a pretrained model $\mathcal{M}$ learned from a source dataset $\mathcal{D}_0 = \{(\mathbf{x}_i^0, y_i^0)\}_{i=1}^{N_0}$, where $\mathbf{x}_i^0 \in \mathcal{X}$ are inputs and $y_i^0 \in \mathcal{Y}_0$ are labels from the source label space. During downstream adaptation, we are given a support set $\mathcal{D}_{train} = \{(\mathbf{x}_j^s, y_j^s)\}_{j=1}^{CK}$ with novel classes $y_j^s \in \mathcal{Y}_{novel}$, where typically $\mathcal{Y}_0 \cap \mathcal{Y}_{novel} = \emptyset$ in strict few-shot learning. The objective is to estimate the posterior distribution over novel classes given a test input $\mathbf{x}$ and information sources:

$$P(y | \mathbf{x}, \mathcal{D}_0, \mathcal{D}_{train}), \quad y \in \mathcal{Y}_{novel}. \quad (2)$$

**Assumption 1 (Conditional Independence of Information Sources).** We assume that the pretraining dataset $\mathcal{D}_0$ and the downstream support set $\mathcal{D}_{train}$ provide conditionally independent evidence about the label $y$ when conditioned on the input $\mathbf{x}$. Formally:

$$P(\mathcal{D}_0, \mathcal{D}_{train} | y, \mathbf{x}) = P(\mathcal{D}_0 | y, \mathbf{x}) \cdot P(\mathcal{D}_{train} | y, \mathbf{x}). \quad (3)$$

This assumption captures the intuition that $\mathcal{D}_0$ and $\mathcal{D}_{train}$ come from different domains/tasks and thus provide complementary, independent information when predicting $y$. While this is an approximation, it provides a tractable starting point for analysis. For a rigorous justification of Assumption 1, please refer to Appendix B.

**Lemma 1 (Exact Factorization).** Under Assumption 1, the joint posterior factorizes as:

$$P(y | \mathbf{x}, \mathcal{D}_0, \mathcal{D}_{train}) = \frac{1}{Z(\mathbf{x})} \cdot \frac{P(y | \mathbf{x}, \mathcal{D}_0) \cdot P(y | \mathbf{x}, \mathcal{D}_{train})}{P(y | \mathbf{x})}, \quad (4)$$

where $Z(\mathbf{x}) = \sum_{y'} \frac{P(y' | \mathbf{x}, \mathcal{D}_0) \cdot P(y' | \mathbf{x}, \mathcal{D}_{train})}{P(y' | \mathbf{x})}$.

*Proof:* Please see Appendix A for the proof.

The exact factorization requires estimating $P(y | \mathbf{x})$, the unconditional prior over labels given the input. In few-shot learning, this is problematic because: (1) $P(y | \mathbf{x})$ is inherently task-dependent and unknown for novel classes. (2) Estimating it reliably would require abundant data, contradicting the few-shot premise. Therefore, we make a simplifying assumption by the transfer learning setting:

**Assumption 2 (Weakly Informative Prior).** For novel classes in few-shot adaptation, the unconditional prior $P(y | \mathbf{x})$ carries little discriminative information compared to the conditioned distributions $P(y | \mathbf{x}, \mathcal{D}_0)$ and $P(y | \mathbf{x}, \mathcal{D}_{train})$. In the absence of strong prior knowledge, we approximate it as uniform over the candidate classes:

$$P(y | \mathbf{x}) \approx \frac{1}{|\mathcal{Y}_{novel}|}, \quad \forall y \in \mathcal{Y}_{novel}. \quad (5)$$

This uniform prior is standard for novel classes with no prior knowledge.

Under Assumption 2, $P(y|\mathbf{x})$ becomes a constant factor that can be absorbed into the normalization:

$$P(y|\mathbf{x}, \mathcal{D}_0, \mathcal{D}_{\text{train}}) \propto P(y|\mathbf{x}, \mathcal{D}_0) \cdot P(y|\mathbf{x}, \mathcal{D}_{\text{train}}). \quad (6)$$

**Lemma 2 (Practical Factorization).** Under Assumptions 1 and 2, the joint posterior is approximately proportional to the product of the individual posteriors:

$$P(y|\mathbf{x}, \mathcal{D}_0, \mathcal{D}_{\text{train}}) \approx \frac{1}{\tilde{Z}(\mathbf{x})} \cdot P(y|\mathbf{x}, \mathcal{D}_0) \cdot P(y|\mathbf{x}, \mathcal{D}_{\text{train}}),$$
$$(7)$$

where $\tilde{Z}(\mathbf{x}) = \sum_{y'} P(y'|\mathbf{x}, \mathcal{D}_0) \cdot P(y'|\mathbf{x}, \mathcal{D}_{\text{train}})$.

**Assumption 3 (Pretrained Model as Approximate Posterior).** A well-calibrated pretrained model $f_\theta(\cdot)$, fine-tuned to minimize cross-entropy on $\mathcal{D}_0$, approximates the Bayesian posterior over the source label space:

$$f_\theta(\mathbf{x})_y \approx P(y|\mathbf{x}, \mathcal{D}_0). \quad (8)$$

For novel classes $y \in \mathcal{Y}_{\text{novel}}$, we assume the model can generalize through zero-shot or few-shot adaptation mechanisms (e.g., via prompt engineering in VLMs).

**Assumption 4 (Support-Set Module as Likelihood Estimator).** A support-set module $g_\phi(\cdot)$ trained on $\mathcal{D}_{\text{train}}$ approximates the posterior given only the few-shot evidence:

$$g_\phi(\mathbf{x})_y \approx P(y|\mathbf{x}, \mathcal{D}_{\text{train}}), \quad y \in \mathcal{Y}_{\text{novel}}. \quad (9)$$

Applying these approximations to Lemma 2, we have:

**Design Principle 1 (Additive Logit Integration).**

$$\log P(y|\mathbf{x}, \mathcal{D}_0, \mathcal{D}_{\text{train}}) \approx \log f_\theta(\mathbf{x})_y + \log g_\phi(\mathbf{x})_y + \text{constant}. \quad (10)$$

The Principle 1 offers clear and actionable design guidelines for building few-shot learning framework: (1) The pretrained model $f_\theta(\cdot)$ and the support-set module $g_\phi(\cdot)$ can be designed and optimized independently. (2) The optimal prediction is a log-linear combination of the outputs from both modules-specifically, the sum of their logits.

**Discussion of Assumptions:** While the conditional independence assumption is an approximation, it provides a principled guideline for decoupling the two information sources. In practice, we still allow limited interaction through joint optimization, which can capture residual dependencies.

The proposed guidelines naturally accommodate and could explain a wide range of existing few-shot learning approaches, for example,

- **Prototypical Networks** (Snell et al., 2017) compute distances between embedded queries and class prototypes from the support set. This process essentially

implements a specialized form of $g_\phi(\cdot)$ that encodes support-set information through metric learning.

- **LDC** (Li et al., 2025) learns to reduce inter-class confusion in logit space via a residual structure, effectively performing a joint refinement of $f_\theta(\cdot)$ and $g_\phi(\cdot)$ through collaborative optimization.

- **AMU-Tuning** (Tang et al., 2024) learns a logit bias to adapt CLIP-based models in few-shot settings, which corresponds to jointly adjusting $f_\theta(\cdot)$ and $g_\phi(\cdot)$.

- **Tip-Adapter** (Zhang et al., 2022) augments a frozen CLIP model with a non-parametric adapter whose predictions are combined with the original model's outputs via a residual connection. This design can be viewed as instantiating an efficient $g_\phi(\cdot)$ module.

Overall, our theoretical analysis provides a general framework and a thorough interpretation of several established few-shot learning methods.

### 3.3. The Bayesian-inspired Optimal Integration Framework(BOIF)

The analysis in Section 3.2 is model-agnostic and various pretrained vision models (e.g., ResNet, ViT) can serve as $f_\theta(\cdot)$. However, to fully leverage the prior $f_\theta(\cdot)$, especially on novel classes, the pretrained model must possess strong generalization capabilities in open-world settings. In this work, we adopt Contrastive Language-Image Pre-training (CLIP) (Radford et al., 2021) as the foundation, as it aligns visual and semantic representations, enabling meaningful priors for unseen categories–unlike classical supervised models constrained to predefined label sets.

Our framework implements a dual-path architecture that realizes $f_\theta(\cdot)$ and $g_\phi(\cdot)$, as illustrated in Fig. 1. In the following part, we first briefly review CLIP and then describe our realization of the two modules.

**3.3.1 Preliminaries of CLIP:** CLIP consists of two parallel encoders: *(a) Visual Encoder ($\mathcal{E}_I$):* Given an input image $\mathbf{x}_i$, it extracts a $d$-dimensional feature vector $\mathbf{z}_i = \mathcal{E}_I(\mathbf{x}_i) \in \mathbb{R}^d$. *(b) Textual Encoder ($\mathcal{E}_T$):* For each category $c \in \{1, \ldots, C\}$, a prompt template (e.g., "a photo of a [CLASS]") is filled with the class name. $\mathcal{E}_T$ projects these prompts into class-specific embeddings $\mathbf{w}_c$, forming a textual classifier matrix $\mathbf{W} = [\mathbf{w}_1, \ldots, \mathbf{w}_C] \in \mathbb{R}^{d \times C}$.

**Zero-Shot Inference:** The zero-shot logits $o_i^{zs}$ of CLIP are computed as the cosine similarity between the visual and textual embeddings, scaled by a temperature $\tau$:

$$o_i^{zs} = \frac{1}{\tau} \mathbf{W}^\top \mathbf{z}_i. \quad (11)$$

The prediction probability is $P(y|\mathbf{x}_i) = \text{Softmax}(o_i^{zs})$.

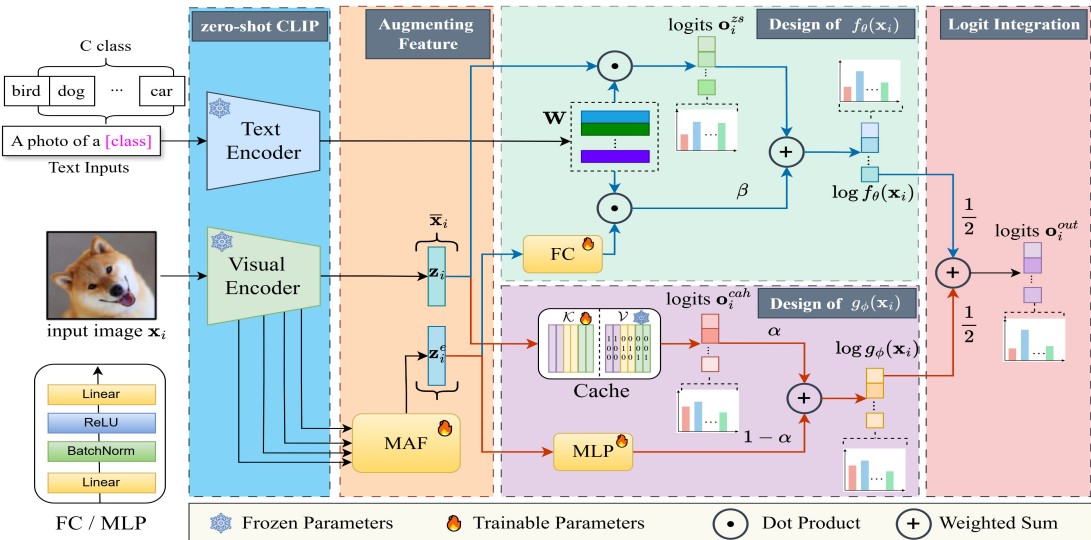

*Figure 1.* **Overview of the proposed Bayesian-inspired Integration framework.** The architecture utilizes a frozen CLIP backbone to extract features. The framework consists of two independent paths: $f_\theta(\mathbf{x}_i)$ generates the refined prior logits; $g_\phi(\mathbf{x}_i)$ derives the support-set posterior logits. The FC (Feature Converter) module projects the enriched multi-level feature $\mathbf{z}_i^e$ into the semantic-aligned space defined by CLIP's text embeddings $\mathbf{W}$, and the MLP module directly maps it to classification logits.

**Connection to Bayesian Framework:** Based on assumption 2, the zero-shot distribution provided by CLIP serves as a well-calibrated approximation of the prior probability $P(y|\mathbf{x}, \mathcal{D}_0)$. However, this prior may be suboptimal in downstream few-shot tasks. Our proposed method aims to fine-tune this prior ($f_\theta(\cdot)$) and designs a support-set-aware module $g_\phi(\cdot)$ operating on the low-dimensional feature $\mathbf{z}_i$.

**3.3.2 Augmenting Feature Representation of $\mathbf{x}_i$:** To mitigate the challenges posed by high-dimensional input data $\mathbf{x}_i$, we use its low-dimensional embedding $\mathbf{z}_i$ extracted from CLIP's visual encoder. However, since standard CLIP focuses on semantic alignment between modalities, it may discard fine-grained spatial and textural details that are critical for discriminative few-shot learning (Xie et al., 2025). Consequently, $\mathbf{z}_i$ alone may be insufficient to represent $\mathbf{x}_i$.

To address this limitation, we integrate a Multi-level Adapter Fusion (MAF) module (Li et al., 2025), which extracts and fuses intermediate feature maps from multiple layers of the CLIP backbone, yielding an enriched representation $\mathbf{z}_i^e$. By combining the original embedding $\mathbf{z}_i$ and the enriched multi-level features $\mathbf{z}_i^e$, we obtain a more comprehensive and expressive feature vector $\bar{\mathbf{x}}_i = \{\mathbf{z}_i, \mathbf{z}_i^e\}$.

**3.3.3 Design of $f_\theta(\cdot)$:** The zero-shot logits $o_i^{zs}$ (Eq. 11) produced by standard CLIP rely solely on the visual embedding $\mathbf{z}_i$. Since $\mathbf{z}_i$ alone may be insufficient to represent the visual image $\mathbf{x}_i$ for discrimination in few-shot tasks, we incorporate the enriched multi-level feature representation $\mathbf{z}_i^e$ obtained via MAF. The composite representation $\bar{\mathbf{x}}_i$ thus serves as a more expressive representation than $\mathbf{z}_i$.

To effectively integrate $\mathbf{z}_i^e$ into the classification logits, we first project it into the semantic-aligned space defined by CLIP's text embeddings $\mathbf{w}_c$. This alignment is performed using an MLP as the feature convertor, denoted as $FC(\mathbf{z}_i^e)$. The logit design of $f_\theta(\cdot)$ is defined as

$$\log f_\theta(\mathbf{x}_i) = o_i^{zs} + \beta \cdot \mathbf{W}^\top \cdot \text{FC}(\mathbf{z}_i^e), \qquad (12)$$

where $\beta \geq 0$ is a tunable weighting parameter that controls the influence of the enriched features. Although the refinement of $f_\theta(\cdot)$ uses support-set data, this is a practical compromise: a frozen pretrained model yields only an approximation of the true Bayesian posterior, and we use support-set to improve the performance of pretrained model. When $\beta = 0$, the model reduces to the original zero-shot CLIP. For $\beta > 0$, the augmented logit term can compensate for representation gaps in the visual encoder, thereby refining the prediction of pretrained model CLIP for downstream few-shot tasks.

**3.3.4 Design of $g_\phi(\cdot)$:** The module $g_\phi(\cdot)$ depends solely on the support set. Following the cache-based design in prior work (Zhang et al., 2022), we implement $g_\phi(\cdot)$ as a similarity-driven cache variant. Given the support set $\mathcal{D}_{train} = \{(\mathbf{x}_i, y_i)\}_{i=1}^{CK}$, we first extract its embedding representation $\{(\mathbf{z}_i, y_i)\}_{i=1}^{CK}$, $\mathbf{z}_i \in \mathbb{R}^{d \times 1}$. Stacking these embeddings yields a cache matrix $\mathcal{K} \in \mathbb{R}^{d \times CK}$ and a corresponding one-hot label matrix $\mathcal{V} \in \mathbb{R}^{C \times CK}$. During training, $\mathcal{K}$ is treated as learnable parameters initialized with the support embeddings, while $\mathcal{V}$ remains fixed.

For an input $\mathbf{x}_i$($y_i$ is unknown at inference), the prediction

is derived from its similarity to the cached support samples:

$$o_i^{cah} = \mathcal{V} \cdot \tanh(\gamma \cdot \mathcal{K}^\top \mathbf{z}_i + \mu), \tag{13}$$

where $\gamma > 0$ and $\mu$ are hyperparameters. The inner product $\mathcal{K}^\top \mathbf{z}_i$ measures cosine-based similarity between the input and each cached support embedding. The output $o_i^{cah}$ thus amounts to a similarity-weighted aggregation of support-set labels. The $\tanh(\cdot)$ function is used to keep the scores within a bounded range, aiding numerical stability.

Recall that our feature representation for $\mathbf{x}_i$ is the composite vector $\bar{\mathbf{x}}_i = \{\mathbf{z}_i, \mathbf{z}_i^e\}$. While $o_i^{cah}$ only exploits $\mathbf{z}_i$, we propose to incorporate the enriched feature $\mathbf{z}_i^e$ through a lightweight MLP to fully leverage the multi-level information. We design the logit of $g_\phi(\cdot)$ as:

$$\log g_\phi(\mathbf{x}_i) = \alpha \cdot o_i^{cah} + (1 - \alpha) \cdot \text{MLP}(\mathbf{z}_i^e), \tag{14}$$

where $\alpha \in [0, 1]$ is a tunable balancing coefficient that controls the relative contribution of the cache-based similarity ($\mathbf{z}_i$) and the enriched multi-level features ($\mathbf{z}_i^e$).

**3.3.5 Logit Integration and Inference:** Based on the additive principle in Eq. 10, we formulate the output logit $o_i^{out}$ as average of the two designed modules:

$$o_i^{out} = 0.5 \left( \log f_\theta(\mathbf{x}_i) + \log g_\phi(\mathbf{x}_i) \right). \tag{15}$$

The final label prediction is obtained by applying the $Argmax$ operation:

$$\hat{y}_i = argmax_j \{o_{i,j}^{out}\}, \tag{16}$$

where $o_{i,j}^{out}$ is the $j-$th entry of output logit vector $o_i^{out}$.

### 3.4. Optimization Objective

Our framework requires optimizing the parameters of the Cache module, the MAF module, the feature converter (FC), and the MLP. The main training objective is the cross-entropy loss $\mathcal{L}_{CE}$ (Eq. 1).Moreover, Eq. 10 shows that $f_\theta$ and $g_\phi$ operate as independent modules, each capturing knowledge from distinct sources–the pretraining dataset and the support set, respectively. To encourage this structural separation and independence, we introduce auxiliary cross-entropy losses applied to each module individually:

$$\mathcal{L}_{aux} = \mathcal{L}_{CE}^{(f)} + \mathcal{L}_{CE}^{(g)}, \tag{17}$$

where $\mathcal{L}_{CE}^{(f)}$ and $\mathcal{L}_{CE}^{(g)}$ denote the cross-entropy losses derived from the predictions $f_\theta(\mathbf{x}_i)$ and $g_\phi(\mathbf{x}_i)$, respectively. In few-shot fine-tuning, a common implicit assumption is that the pretrained model's behavior should not be drastically altered, and outputs should remain relatively stable. To preserve this assumption, we impose an L1 consistency penalty that discourages large deviations from this anchor:

$$\mathcal{L}_{pcr} = \| \log f_\theta(\mathbf{x}_i) - o_i^{zs} \|_1 + \| \log g_\phi(\mathbf{x}_i) - o_i^{zs} \|_1. \tag{18}$$

The overall objective function is:

$$\mathcal{L} = \mathcal{L}_{CE} + \lambda_1 \mathcal{L}_{aux} + \lambda_2 \mathcal{L}_{pcr}, \tag{19}$$

where $\lambda_1$ and $\lambda_2$ are regularization hyperparameters.

**Remark 1(Difference from existing methods):** Although numerous few-shot learning methods based on CLIP exist, our work offers a unified framework and a theoretical interpretation of the few-shot learning. Our method explicitly distinguishes the roles and optimization pathways of $f_\theta(\cdot)$ and $g_\phi(\cdot)$. Moreover, our technical design differs from prior works in several key aspects: we attribute performance limitations partly to feature-representation gaps and accordingly introduce feature representation augmentation; we also propose a simplified yet effective cache module than Tip-Adapter (Zhang et al., 2022).

## 4. Experiments

We compare our method with state-of-the-art approaches on standard few-shot benchmarks and out-of-distribution (OOD) settings. Furthermore, we provide ablation studies to validate the effectiveness of our module design.

### 4.1. Experimental Setup

**Datasets.** We evaluate our method on 11 widely-used image classification datasets: ImageNet (Deng et al., 2009), Caltech101 (Fei-Fei et al., 2004), DTD (Cimpoi et al., 2014), EuroSAT (Helber et al., 2019), FGVCAircraft (Maji et al., 2013), Flowers102 (Nilsback & Zisserman, 2008), Food101 (Bossard et al., 2014), OxfordPets (Parkhi et al., 2012), StanfordCars (Krause et al., 2013), SUN397 (Xiao et al., 2010), and UCF101 (Soomro, 2012). For the few-shot settings, we follow the standard protocol in CLIP (Radford et al., 2021) and CoOp (Zhou et al., 2022b), utilizing 1, 2, 4, 8, and 16 shots per class for training and evaluating on the test sets. Additionally, to assess the domain generalization capability, we train our model on ImageNet (16-shot) and evaluate it on ImageNet-V2 (Recht et al., 2019) and ImageNet-Sketch (Wang et al., 2019) as out-of-distribution (OOD) targets.

**Implementation Details.** For a fair comparison, we utilize ResNet-50 (He et al., 2016) as the image encoder for CLIP unless otherwise specified. We also provide results with ViT-B/16 (Dosovitskiy et al., 2020) as backbone in the OOD setting. During training, both the pre-trained CLIP image and text encoders remain frozen. We only fine-tune the additional modules (i.e., EFC, MLP, and cache-keys). The model is trained for 50 epochs using the AdamW optimizer with an initial learning rate 0.0001 and a batch size 64. A cosine annealing scheduler is applied to adjust the learning rate. For data augmentation, we employ standard random resized cropping and horizontal flipping. Experiments are implemented in PyTorch on a single

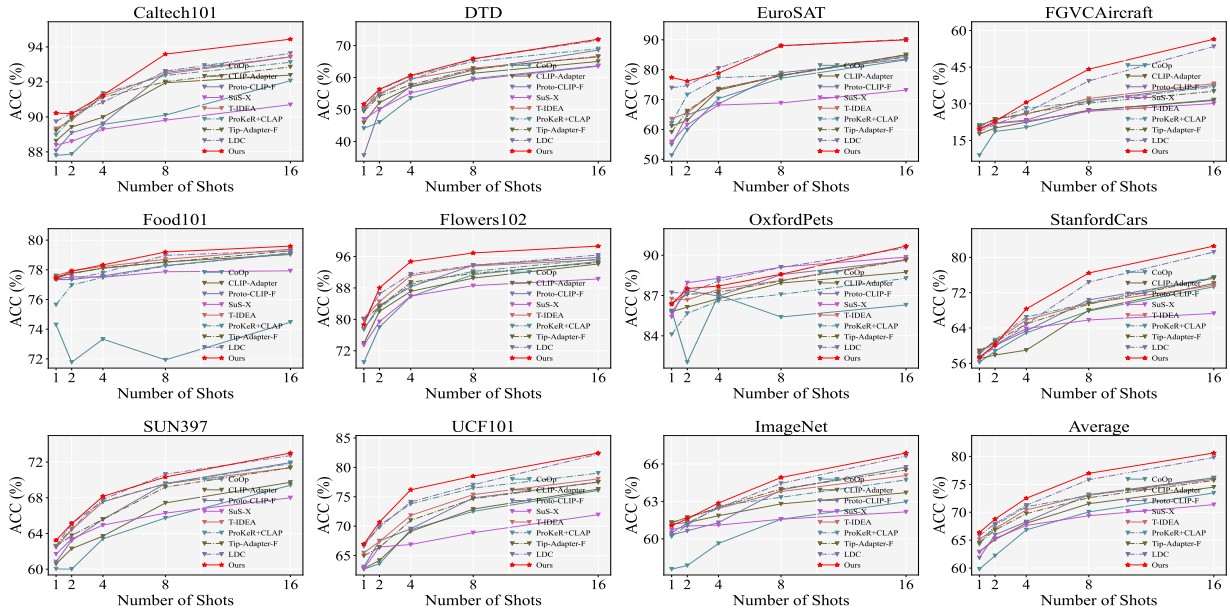

*Figure 2.* Classification accuracy performance comparison on 11 datasets, and the last one is the average performance on these 11 datasets.

NVIDIA RTX3060 GPU. The code is released on https://anonymous.4open.science/r/Anonymous_BOIF-4E3D/.

**Comparison baseline methods.** We compare our approach with state-of-the-art methods, including CLIP baselines (Zero-shot CLIP (Radford et al., 2021), LP-CLIP (Radford et al., 2021)), prompt learning and adapter-based methods (CoOp (Zhou et al., 2022b), CLIP-Adapter (Gao et al., 2024), Tip-Adapter-F (Zhang et al., 2022)), and recent advanced approaches (SuS-X (Udandarao et al., 2023), Proto-CLIP-F (P et al., 2024), ProKeR+CLAP (Bendou et al., 2025), LDC (Li et al., 2025), T-IDEA (Ye et al., 2026)).

### 4.2. Experimental Results

❶ **Accuracy Performance over 11 Datasets.** We begin by comparing our method with baseline approaches across 11 diverse datasets in Fig. 2. Fig. 2 summarizes performance across datasets and shows that our method (red curve) consistently outperforms all competing techniques. Furthermore, Table 1 reports the average accuracy over the 11 datasets, demonstrating that our approach achieves state-of-the-art performance. Specifically, our method attains 66.35% accuracy in the challenging 1-shot setting and 80.61% in the 16-shot setting, consistently outperforming the second-best methods, LDC and Tip-Adapter-F. We further evaluate our method on the ImageNet dataset in Table 4. The results agree well with Table 1. Hence, we arrange these results in Appendix C.1 due to space limitations.

❷ **Robustness to Distribution Shift.** To rigorously assess robustness under distribution shifts, we further incorporate several competitive baselines, including CoCoOp (Zhou

*Table 1.* Accuracy comparison of different methods over 11 datasets. '-' denotes unavailable results. A higher value is better.

| Method | K-Shot Accuracy (%) | | | | | |
|---|---|---|---|---|---|---|
| | 0 | 1 | 2 | 4 | 8 | 16 |
| ZS-CLIP[ICML2021] | 58.87 | - | - | - | - | - |
| LP-CLIP[ICML2021] | - | 36.67 | 47.61 | 57.19 | 64.98 | 71.10 |
| CoOp[IJCV2022] | - | 59.80 | 62.21 | 66.84 | 70.05 | 73.45 |
| Tip-Adapter-F[ECCV2022] | - | 64.55 | 66.79 | 69.76 | 72.59 | 75.69 |
| SuS-X[ICCV2023] | - | 62.87 | 65.29 | 67.64 | 69.37 | 71.36 |
| CLIP-Adapter[IJCV2024] | - | 62.90 | 65.11 | 68.02 | 71.52 | 74.50 |
| Proto-CLIP-F[IROS2024] | - | 61.84 | 65.96 | 68.29 | 73.13 | 76.18 |
| ProKeR+CLAP[CVPR2025] | - | 64.28 | 67.66 | 71.07 | 73.09 | 76.11 |
| LDC[CVPR2025] | - | 65.71 | 67.92 | 71.17 | 75.79 | 79.78 |
| T-IDEA[PR2026] | - | 65.11 | 67.07 | 70.41 | 73.12 | 75.91 |
| Ours | - | 66.35↑ | 68.77↑ | 72.50↑ | 76.96↑ | 80.61↑ |

et al., 2022a), CALIP-FS (Guo et al., 2023), MaPLe (Khattak et al., 2023), RPO (Lee et al., 2023b), ProKeR (Bendou et al., 2025), and MMA (Yang et al., 2024). We first train 16-shot models on ImageNet as the source domain and then evaluate them on ImageNet-V2 and Sketch as target domains. Two CLIP visual backbones, ResNet-50 and ViT-B/16, are employed in the experiments, and the corresponding results are summarized in Table 2.

Using ResNet-50 as the CLIP backbone, our method achieves 66.84% accuracy on the source domain and 58.45% on ImageNet-V2 in Table 2. When switching to the ViT-B/16 backbone, our approach surpasses state-of-the-art methods (RPO and LDC) on both OOD benchmarks, attaining 66.25% on ImageNet-V2 and 49.41% on Sketch.

❸ **Ablation Study.** We perform a comprehensive ablation study to assess the contribution of each component in our

*Table 2.* Comparison of different methods under OOD setting. A higher value is better.

| | Method | Source | Target | |
|---|---|---|---|---|
| | | ImageNet | V2 | Sketch |
| ResNet-50 | ZS-CLIP | 60.33 | 53.27 | 35.44 |
| | LP-CLIP | 56.13 | 45.61 | 19.13 |
| | CoOp | 62.95 | 54.58 | 31.04 |
| | CoCoOp | 62.81 | 55.72 | 34.48 |
| | CALIP-FS | 65.81 | 55.98 | 35.37 |
| | Tip-Adapter | 62.03 | 54.60 | 35.90 |
| | Tip-Adapter-F | 65.51 | 57.11 | 36.00 |
| | CLIP-Adapter | 63.59 | 55.69 | 35.68 |
| | ProKeR | 64.47 | 56.08 | **36.01** |
| | ProKeR+CLAP | 64.72 | 56.12 | 35.32 |
| | LDC | 66.63 | 58.03 | 35.52 |
| | **Ours** | **66.84** | **58.45** | 36.00 |
| ViT-B/16 | ZS-CLIP | 66.73 | 60.83 | 46.15 |
| | CoOp | 71.51 | 64.20 | 47.99 |
| | CoCoOp | 71.02 | 64.07 | 48.75 |
| | MaPLe | 70.72 | 64.07 | 49.15 |
| | MMA | 71.00 | 64.33 | 49.13 |
| | RPO | 71.67 | 65.13 | 49.27 |
| | LDC | 73.88 | 66.10 | 48.85 |
| | **Ours** | **74.34** | **66.25** | **49.41** |

framework, including the Cache module, the MLP, and the Feature Converter (FC). The results, averaged over 11 datasets under the 16-shot setting, are reported in Table 3. For the configuration that uses only the Cache module (Row 2), we follow the same hyperparameter setup as Tip-Adapter (Zhang et al., 2022).

Comparing the baseline (Row 1) with models that incorporate a single enriched feature module (Rows 3 and 4), we observe that adding either the MLP or the FC module substantially improves accuracy on ResNet-50, respectively. These results support our hypothesis that the pretrained CLIP embedding is insufficient for few-shot learning, thereby underscoring the importance of feature augmentation. Furthermore, contrasting the results in Row 6 and Row 8 of Table 3 demonstrates the validity of Corollary 1, confirming that a log-linear combination of the refined prior and the support-set evidence leads to improved performance.

*Table 3.* Effect analysis with/without each module over 11 datasets.

| | W/o modules | | | CLIP Backbone | |
|---|---|---|---|---|---|
| # | Cache($z_i$) | MLP($z_i^e$) | FC($z_i^e$) | ResNet50 | ViT-B/16 |
| 1 | × | × | × | 58.87 | 65.52 |
| 2 | ✓ | × | × | 76.11 | 81.87 |
| 3 | × | ✓ | × | 78.67 | 82.16 |
| 4 | × | × | ✓ | 78.87 | 82.22 |
| 5 | ✓ | × | ✓ | 79.61 | 82.93 |
| 6 | ✓ | ✓ | × | 79.82 | 82.81 |
| 7 | × | ✓ | ✓ | 80.31 | 83.17 |
| 8 | ✓ | ✓ | ✓ | 80.61 | 83.78 |

We additionally conduct correlation analysis of $\log f_\theta(\mathbf{x}_i)$ and $\log g_\phi(\mathbf{x}_i)$ (Appendix C.5) , logit fusion weight sensitivity (Appendix C.4) and cross-backbone generalization tests (Appendix C.6). These experiments collectively verify the rationality of our core assumptions, the optimality of equal-weight fusion, and the generalizability of the proposed Bayesian integration principle.

❹ **Visualization of Feature Distributions.** To assess the effectiveness of our feature augmentation, we perform *t-SNE* (Arora et al., 2018) visualization on 10 randomly selected classes from the StanfordCars (Krause et al., 2013) test set. We compare two types of feature representations from the test samples: (a) the standard visual embedding ($\mathbf{z}_i$) obtained directly from the frozen CLIP visual encoder $\mathcal{E}_I$, and (b) the augmented visual embedding ($\bar{\mathbf{x}}_i^{cat}$). The augmented feature is $\bar{\mathbf{x}}_i = \{\mathbf{z}_i, \mathbf{z}_i^e\}$, and we explicitly concatenate the two components, yielding $\bar{\mathbf{x}}_i^{cat} = \text{Concat}(\mathbf{z}_i, \mathbf{z}_i^e)$. This concatenated representation corresponds to the actual input of our adaptive module $g_\phi(\cdot)$.

Fig. 3 compares the feature distributions of $\mathbf{z}_i$ and $\bar{\mathbf{x}}_i^{cat}$, where each of the 10 classes is marked with a distinct color. In Fig. 3(a), the standard CLIP embedding $\mathbf{z}_i$ exhibits substantial overlap across different classes, suggesting limited discriminative capability for fine-grained classification. In contrast, Fig. 3(b) illustrates the augmented features produced by our BOIF framework. By incorporating the structural information encoded in $\mathbf{z}_i^e$, the augmented representation $\bar{\mathbf{x}}_i^{cat}$ demonstrates markedly improved intra-class compactness and inter-class separability. This visual comparison confirms that the augmented features offer stronger discriminative cues than the original embeddings, thereby validating the effectiveness of our feature augmentation design.

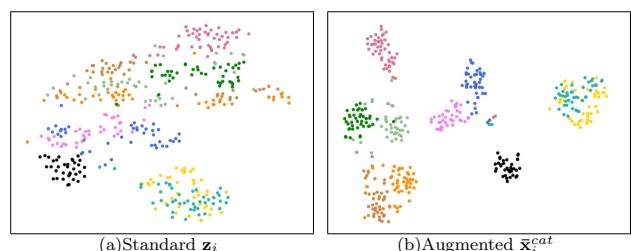

(a)Standard $\mathbf{z}_i$     (b)Augmented $\bar{\mathbf{x}}_i^{cat}$

*Figure 3.* **t-SNE visualization of features on Stanford Cars dataset for 10 random classes(denoted by different colors).** (a) The standard CLIP embeddings have serious class overlap. (b) Our augmented representation $\bar{\mathbf{x}}_i^{cat}$ has better intra-class compactness and inter-class separability.

## 5. Conclusion

In this paper, we have proposed presented a theoretically grounded framework(BOIF) that addresses the fundamental challenge of effectively combining pretrained knowledge with limited support-set evidence in few-shot classification.

We prove that the posterior is achieved through the additive combination of logits from the pretrained prior and the few-shot likelihood. Based on the analysis, we then design a dual-path neural model that uses CLIP alongside a specialized cache-based module. Extensive empirical evaluations across 11 standard benchmarks demonstrate that BOIF not only outperforms state-of-the-art methods but also exhibits superior robustness under out-of-distribution scenarios. By providing an analytical interpretation of the adaptation process, our work bridges the gap between rigorous Bayesian analysis and practical neural architecture design. We believe this framework offers a way for better adaptation strategies in few-shot classification task.

**Limitations**: (a) Theoretical assumptions: The conditional independence assumption, while providing a clean design principle, may not hold strictly. Future work could explore more sophisticated models that capture dependencies. (b) Fixed weighting: Our framework uses equal weighting of prior and likelihood logits. Learning task-adaptive weights (e.g., based on uncertainty) could further improve performance. (c) Computational overhead: The MAF module increases parameters; designing more efficient feature enhancement remains important for deployment.

## Impact Statement

This work presents a significant step forward in the theoretical understanding and practical application of few-shot learning. By providing the adaptation of large-scale foundation models in a Bayesian perspective, we move from heuristic engineering to rigorous, principled network design. Our method is not limited to CLIP in the experiments; other pretrained models could also serve as the backbone of our framework. Our method has broad societal implications, particularly in data-scarce domains, such as rare disease diagnosis in medical imaging and endangered species monitoring in ecology. We believe this framework offers a general way to design better few-shot classification models. By enabling high-performance classification with minimal examples, our framework reduces the resource cost for AI deployment.

## Acknowledgments

The authors acknowledge the financial support from the National Natural Science Foundation of China (Grant Nos. 62476173, 62276171, 62002233, 61972145, 62532007), CCF-Huawei Populus Grove Fund (Grant Nos. CCF-HuaweiFM2024004), the Shenzhen Fundamental Research-General Project (Grant Nos. JCYJ20240813142610014, JCYJ20240813141503005, ZDCY20250901110940006), Guangdong Basic and Applied Basic Research Foundation (Grant Nos. 2024A1515011938 and 2020B1515120028), Major Special Project for Philosophy and Social Sciences Research of the Ministry of Education (Grant No. 2025JZDZ010).

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

## A. Proof of Lemma 1

*Proof.* We start from Bayes' theorem:

$$P(y|\mathbf{x}, \mathcal{D}_0, \mathcal{D}_{\text{train}}) = \frac{P(\mathcal{D}_0, \mathcal{D}_{\text{train}}|y, \mathbf{x})P(y|\mathbf{x})P(\mathbf{x})}{P(\mathcal{D}_0, \mathcal{D}_{\text{train}}, \mathbf{x})}$$
$$\propto P(\mathcal{D}_0, \mathcal{D}_{\text{train}}|y, \mathbf{x})P(y|\mathbf{x}), \tag{20}$$

where the proportionality follows since the denominator $P(\mathcal{D}_0, \mathcal{D}_{\text{train}}|\mathbf{x})$ is independent of $y$.

Applying Assumption 1:

$$P(y|\mathbf{x}, \mathcal{D}_0, \mathcal{D}_{\text{train}}) \propto P(\mathcal{D}_0|y, \mathbf{x}) \cdot P(\mathcal{D}_{\text{train}}|y, \mathbf{x}) \cdot P(y|\mathbf{x}). \tag{21}$$

Now consider the individual terms. Using Bayes' theorem in reverse:

$$P(\mathcal{D}_0|y, \mathbf{x}) = \frac{P(y|\mathbf{x}, \mathcal{D}_0) \cdot P(\mathcal{D}_0|\mathbf{x})}{P(y|\mathbf{x})} \propto \frac{P(y|\mathbf{x}, \mathcal{D}_0)}{P(y|\mathbf{x})},$$
$$P(\mathcal{D}_{\text{train}}|y, \mathbf{x}) = \frac{P(y|\mathbf{x}, \mathcal{D}_{\text{train}}) \cdot P(\mathcal{D}_{\text{train}}|\mathbf{x})}{P(y|\mathbf{x})} \propto \frac{P(y|\mathbf{x}, \mathcal{D}_{\text{train}})}{P(y|\mathbf{x})}, \tag{22}$$

where we drop the terms $P(\mathcal{D}_0|\mathbf{x})$ and $P(\mathcal{D}_{\text{train}}|\mathbf{x})$ as they are independent of $y$.

Substituting back into Eq. 21:

$$P(y|\mathbf{x}, \mathcal{D}_0, \mathcal{D}_{\text{train}}) \propto \frac{P(y|\mathbf{x}, \mathcal{D}_0)}{P(y|\mathbf{x})} \cdot \frac{P(y|\mathbf{x}, \mathcal{D}_{\text{train}})}{P(y|\mathbf{x})} \cdot P(y|\mathbf{x})$$
$$= \frac{P(y|\mathbf{x}, \mathcal{D}_0) \cdot P(y|\mathbf{x}, \mathcal{D}_{\text{train}})}{P(y|\mathbf{x})}, \tag{23}$$

which completes the proof. $\square$

## B. Justification of Conditional Independence Assumption

One concern about the consumption is that $\mathcal{D}_0$ and $\mathcal{D}_{\text{train}}$ are not independent, and the two datasets are drawn from the same visual world. Here, we use the latent variable model to validate the assumption. In latent variable models (Bartholomew et al., 2011), observed data are typically assumed to be generated from latent variables $z$, i.e., $P((x, y)|z)$. Different datasets may share the same visual world, but reflect the visual world from different viewpoints. $z$ represents the latent variables of real world, and hence different datasets only depend on $z$. Consequently, both the training set and support set depend only on $z$, giving

$$P(\mathcal{D}_0, \mathcal{D}_{\text{train}}|z) = P(\mathcal{D}_0|z) \cdot P(\mathcal{D}_{\text{train}}|z). \tag{24}$$

Under the common point estimation, we have

$$P(\mathcal{D}_0, \mathcal{D}_{train}|(x, y)) = P(\mathcal{D}_0, \mathcal{D}_{train}|z)P(z|(x, y))$$
$$= P(\mathcal{D}_0|z) \cdot P(\mathcal{D}_{\text{train}}|z)P(z|(x, y))$$
$$= [P(\mathcal{D}_0|z)P(z|(x, y))] \cdot [P(\mathcal{D}_{\text{train}}|z)P(z|(x, y))]/P(z|(x, y))$$
$$= P(\mathcal{D}_0|(x, y)) \cdot P(\mathcal{D}_{\text{train}}|(x, y))/P(z|(x, y))$$
$$\propto P(\mathcal{D}_0|(y, x)) \cdot P(\mathcal{D}_{\text{train}}|(y, x)). \tag{25}$$

Thus, the assumption holds even when the two datasets are drawn from the same visual world.

## C. Additional Experiments

### C.1. Accuracy Performance on ImageNet

We further evaluate our method on the ImageNet dataset under different shot settings in Table 4. Our approach substantially outperforms state-of-the-art baselines when $K \geq 4$. In particular, it achieves a peak accuracy of 66.84% in the 16-shot setting, exceeding the strong competitors LDC (66.63%) and Tip-Adapter-F (65.51%).

*Table 4.* Comparison of different methods on ImageNet.

| Method | K-Shot Accuracy (%) | | | | | |
|---|---|---|---|---|---|---|
| | 0 | 1 | 2 | 4 | 8 | 16 |
| ZS-CLIP$^{ICML2021}$ | 60.34 | - | - | - | - | - |
| LP-CLIP$^{ICML2021}$ | - | 22.17 | 31.90 | 41.20 | 49.52 | 56.13 |
| CoOp$^{IJCV2022}$ | - | 57.15 | 57.81 | 59.99 | 61.56 | 62.95 |
| Tip-Adapter$^{ECCV2022}$ | - | 60.70 | 60.92 | 60.95 | 61.48 | 62.00 |
| Tip-Adapter-F$^{ECCV2022}$ | - | **61.32** | 61.69 | 62.52 | 64.00 | 65.51 |
| SuS-X$^{ICCV2023}$ | - | 60.73 | 61.03 | 61.10 | 61.57 | 62.16 |
| CLIP-Adapter$^{IJCV2024}$ | - | 61.20 | 61.52 | 61.84 | 62.68 | 63.59 |
| Proto-CLIP-F$^{IROS2024}$ | - | 60.32 | 60.64 | 61.30 | 63.92 | 65.75 |
| ProKeR$^{CVPR2025}$ | - | 60.59 | 61.09 | 62.05 | 62.86 | 64.54 |
| ProKeR+CLAP$^{CVPR2025}$ | - | 60.19 | 61.11 | 62.64 | 63.33 | 64.72 |
| LDC$^{CVPR2025}$ | - | 60.48 | 61.25 | 62.47 | 64.44 | 66.63 |
| T-IDEA$^{PR2026}$ | - | 61.28 | **61.73** | 62.43 | 63.79 | 65.08 |
| Ours | - | 61.01 | 61.57 | **62.87** | **64.92** | **66.84** |

*Table 5.* Efficiency comparison with other existing methods on 16-shot ImageNet

| Method | Training time (one epoch) | GFLOPs | Tunable Parameters | Accuracy |
|---|---|---|---|---|
| CoOp | 269.91 s | 2954.67 | 0.0082 M | 62.95 |
| CLIP-Adapter | 9.93 s | 6.14 | 0.5243 M | 63.59 |
| Tip-Adapter-F | 11.77 s | 6.17 | 16.3840 M | 65.51 |
| LDC | 27.07 s | 9.99 | 17.2554 M | 66.63 |
| Ours | 27.77 s | 10.02 | 28.44 M | 66.84 |

## C.2. Efficiency and Computational Complexity

We evaluate the computational efficiency of our framework on an NVIDIA RTX 3090 GPU, with results summarized in Table 5. Compared to CoOp, our method achieves a significant accuracy gain (66.84% vs. 62.95%) while being substantially more efficient, requiring over $200\times$ fewer GFLOPs and $10\times$ less training time per epoch. When compared to the strong LDC baseline, our approach exhibits nearly identical training latency (27.77s) and computational complexity (10.02 GFLOPs) while attaining superior performance. Although the design of our method increases the tunable parameter count to 28.44M, the overall computational overhead remains highly practical for real-world few-shot adaptation.

## C.3. Evaluation on Out-of-Distribution Scenarios

We extend OOD evaluation to ImageNet-A (Hendrycks et al., 2021b) and ImageNet-R (Hendrycks et al., 2021a). Table 6 reports results with ResNet-50 backbone. BOIF achieves the highest average accuracy across all OOD targets, outperforming LDC and other baselines.

*Table 6.* Comparison of different methods under OOD setting.

| Method | Source | Target | | | | |
|---|---|---|---|---|---|---|
| | ImageNet | -V2 | -Sketch | -A | -R | Average |
| ZS-CLIP | 60.33 | 53.27 | 35.44 | 21.65 | 56.00 | 41.59 |
| CoOp | 62.95 | 54.58 | 31.04 | 22.12 | 54.96 | 40.68 |
| Tip-Adapter | 61.43 | 54.13 | 35.71 | **23.63** | 60.41 | 43.47 |
| Tip-Adapter-F | 65.51 | 57.11 | **36.00** | 22.33 | 59.63 | 43.77 |
| CALIP-FS | 65.81 | 55.98 | 35.37 | 23.42 | 56.74 | 42.88 |
| LDC | 66.63 | 58.03 | 35.52 | 22.60 | 60.28 | 44.11 |
| **Ours** | **66.84** | **58.45** | **36.00** | 22.93 | **60.50** | **44.47** |

## C.4. Sensitivity Analysis of Logit Fusion Weight

To validate the theoretical optimality of our equal-weight logit fusion strategy, we perform a sensitivity analysis on the balancing coefficient $\rho$:

$$o^{\text{out}} = (1 - \rho)\log f_\theta + \rho \log g_\phi.$$

As illustrated in Figure 4, we evaluate the 16-shot average accuracy over 11 datasets with varying $\rho \in [0.3, 0.7]$. The performance clearly peaks at $\rho = 0.5$, which strongly confirms our theoretically derived equal-weight fusion rule. This demonstrates that our fusion strategy is not a heuristic combination of existing components, but a principled and empirically optimal design. The optimal performance is consistently achieved at $\rho = 0.5$, validating our Bayes-optimal equal-weight logit fusion.

## C.5. Correlation Analysis

To empirically validate Assumption 1, We have performed a correlation analysis. Specifically, after training with 4-shot support sets, we computed the Pearson correlation between the logits of the two pathways across all test samples. The overall average correlation is 0.34. Moreover, the correlation distribution is shown in Table 7. The results indicate weak correlation of the two pathways' logits, providing empirical support for our conditional independence assumption. Furthermore, prior work (Mayilvahanan et al., 2024) has shown that the generalization ability of CLIP stems from effective feature extraction rather than train-test similarity, suggesting that the shared visual world does not necessarily imply strong dependence between the two information sources.

*Table 7.* Pearson correlation distribution between the two pathways' logits.

| Range | 0–0.1 | 0.1–0.2 | 0.2–0.3 | 0.3–0.4 | 0.4–0.5 | 0.5–0.6 | 0.6–0.7 | 0.7–0.8 | 0.8–0.9 | 0.9–1.0 |
|---|---|---|---|---|---|---|---|---|---|---|
| Fraction of samples (%) | 4.5 | 13.6 | **23.4** | 22.1 | 15.3 | 6.0 | 1.5 | 1.6 | 3.6 | 3.3 |

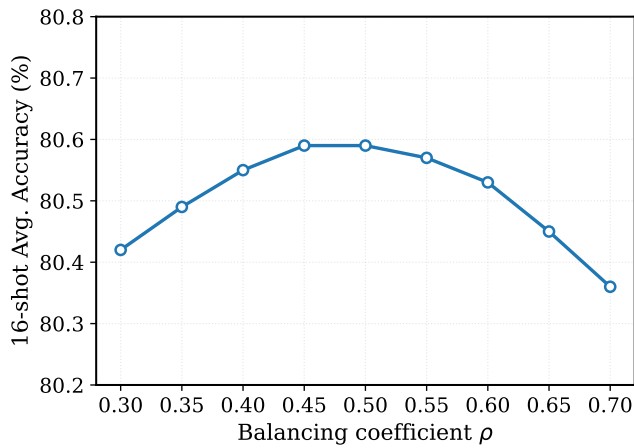

*Figure 4.* Average classification accuracy with different balancing coefficients $\rho$.

## C.6. Generalization to Vision-Only Pre-trained Models

To demonstrate the general applicability of our proposed Bayesian integration principle, we conduct experiments on a vision-only pre-trained model, DINOv2, instead of CLIP. Since DINOv2 lacks a text encoder for semantic alignment, we use the class prototype (mean of support set features extracted by DINOv2 for each class) as the replacement for CLIP's text embeddings. We evaluate performance on the same 11 benchmark datasets.

As shown in Table 8, our method still outperforms all baselines across all few-shot settings, even on vision-only pre-trained models. This validates that our core contribution—the Bayes-optimal logit fusion principle—is model-agnostic and generalizable beyond vision-language models like CLIP.

*Table 8.* Average few-shot classification accuracy over 11 datasets using DINOv2 as the backbone. A higher value is better.

| Method | K-Shot Accuracy (%) | | | | |
|---|---|---|---|---|---|
| | 1 | 2 | 4 | 8 | 16 |
| DINOv2+prototype | 56.84 | 63.03 | 68.10 | 72.52 | 74.59 |
| LP-DINOv2 | 56.81 | 63.14 | 68.91 | 74.60 | 79.09 |
| Tip-Adapter-F | 57.37 | 63.60 | 69.87 | 76.79 | 81.86 |
| Ours | **57.77** | **63.81** | **70.46** | **77.52** | **82.40** |

