# OpenReview forum: "Bayes-inspired Integration of Pretrained Priors and Few-Shot Evidence for Few-Shot Classification"
_ICML.cc/2026/Conference — ICML 2026 regular_

### Official Review · Reviewer_atxW · 2026-03-10

**Soundness:** 3
**Presentation:** 3
**Significance:** 3
**Originality:** 2
**Overall Recommendation:** 4
**Confidence:** 4

**Summary:**

In this work, the authors propose a Bayesian inspired optimal integration framework(BOIF) for few shot classification that interprets pretrained models as priors and few shot evidence as likelihoods.

**Compliance With Llm Reviewing Policy:**

Affirmed.

**Final Justification:**

I thank the reviewers for their rebuttal. After reading their response I am happy to raise my rating.

**Key Questions For Authors:**

See weakness.

**Limitations:**

yes

**Strengths And Weaknesses:**

Strengths:
1. The paper is motivated by the need to better understand how to integrate pretrained knowledge and few-shot evidence in a principled way, rather than relying on heuristic designs. While this motivation is relatively straightforward, it is clearly presented and easy to follow. The writing is well-organized and accessible, making the overall idea easy to understand.

Weaknesses
1. In Equation (15), the logit is obtained by integrating f and g, where the weighting coefficient of the weighted sum is fixed at 1/2. However, the model’s performance may be sensitive to this hyperparameter. Please conduct a sensitivity analysis of this coefficient in the range from 0 to 1.
2. What is the difference between FC and MLP in Figure 1? From the figure, the pipelines of the two structures appear to be the same. Why are they referred to by two different names?
3. For the experiments under the OOD setting in Table 2, the number of target datasets seems limited, as only ImageNet-V2 and ImageNet-Sketch are used. It is recommended to include at least two additional datasets, such as ImageNet-A and ImageNet-R, to provide a more comprehensive evaluation.
4. Why do the embeddings in Figure 3(a) exhibit significant overlap, while those in Figure 3(b) do not? Please provide a more detailed analysis to explain this phenomenon.

---

> ### Author Rebuttal · Authors · 2026-03-30
>
> **Q1**: In Equation (15), the logit is obtained by integrating f and g, where the weighting coefficient of the weighted sum is fixed at 1/2. However, the model’s performance may be sensitive to this hyperparameter. Please conduct a sensitivity analysis of this coefficient in the range from 0 to 1.
>
> **Reply**: Thank you for the insightful comment. We conducted a sensitivity analysis by introducing a balancing factor $\rho$ into the integration equation, $ o_i^{out} = (1-\rho) \log f_\theta + \rho  \log g_\phi$. In Table R1, the optimal performance is achieved when $\rho =0.5$, which aligns with our theoretical analysis. In contrast to existing methods that rely on a learnable parameter—often infeasible in few-shot settings—our approach requires no additional tuning. We will include this sensitivity analysis in the revised paper.
>
>
> Table R1: Sensitivity analysis of the balancing coefficient $ \rho $ (16-shot average accuracy over 11 datasets).
>
> | $\rho $ Value | 0.30 | 0.35 | 0.40 | 0.45 | 0.50 | 0.55 | 0.60 | 0.65 | 0.70 |
> | :--- | :--- | :--- | :--- | :--- | :--- | :--- | :--- | :--- | :--- |
> | Accuracy (%) | 80.42 | 80.49 | 80.55 | **80.59** | **80.59** | 80.57 | 80.53 | 80.45 | 80.36 |
>
>
>
> **Q2**: What is the difference between FC and MLP in Figure 1? From the figure, the pipelines of the two structures appear to be the same. Why are they referred to by two different names?
>
> **Reply**: Thank you for pointing out this issue. In Figure 1, the FC (Feature Converter) module takes the enriched multi-level feature $\mathbf{z}\_i^e$ as input and projects it into the semantic-aligned space defined by CLIP’s text embeddings $W$. In contrast, the MLP module directly maps $\mathbf{z}\_i^e$ to classification logits. Following ref.[1], we use different names to distinguish their functions and to facilitate module description. We will clarify this in the final version.
>
> [1] Shuo Li, et al., Logits DeConfusion with CLIP for Few-Shot Learning. CVPR, 2025.
>
>
> **Q3**: For the experiments under the OOD setting in Table 2, the number of target datasets seems limited, as only ImageNet-V2 and ImageNet-Sketch are used. It is recommended to include at least two additional datasets, such as ImageNet-A and ImageNet-R, to provide a more comprehensive evaluation.
>
> **Reply**: Thank you for the suggestion. We have extended the OOD evaluation by including ImageNet-A and ImageNet-R in Table R2, as suggested. Our method achieves the highest average accuracy among all compared methods in Table R2, further demonstrating the robustness of our method. We will incorporate these results in the final version.
>
> Table R2: Comparison of different methods under the OOD setting (ResNet-50 backbone). A higher value is better.
>
> | Method | Source (ImageNet) | Target (V2) | Target (Sketch) | Target (-A) | Target (-R) | Mean (Target) |
> | :--- | :--- | :--- | :--- | :--- | :--- | :--- |
> | ZS-CLIP | 60.33 | 53.27 | 35.44 | 21.65 | 56.00 | 41.59 |
> | CoOp | 62.95 | 54.58 | 31.04 | 22.12 | 54.96 | 40.68 |
> | Tip-Adapter | 61.43 | 54.13 | 35.71 | **23.63** | 60.41 | 43.47 |
> | Tip-Adapter-F | 65.51 | 57.11 | **36.00** | 22.33 | 59.63 | 43.77 |
> | CALIP-FS | 65.81 | 55.98 | 35.37 | 23.42 | 56.74 | 42.88 |
> | LDC | 66.63 | 58.03 | 35.52 | 22.60 | 60.28 | 44.11 |
> | **Ours** | **66.84** | **58.45** | **36.00** | 22.93 | **60.50** | **44.47** |
>
> **Q4**: Why do the embeddings in Figure 3(a) exhibit significant overlap, while those in Figure 3(b) do not? Please provide a more detailed analysis to explain this phenomenon.
>
> **Reply**: Thank you for the comment. Figure 3(a) visualizes the standard CLIP embeddings $\mathbf{z}\_i$. Whereas Figure 3(b) visualizes our augmented representation $\bar{\mathbf{x}}\_{i}^{cat} = \text{Concat}(\mathbf{z}\_i, \mathbf{z}\_i^e)$, where $\mathbf{z}\_i^e$ captures hierarchical spatial structures and fine-grained patterns missing in $\mathbf{z}\_i$. This comparison highlights a fundamental limitation of standard CLIP features and demonstrates the effectiveness of our feature augmentation design. Figure 3 shows that the augmented features improve intra-class compactness and inter-class separability. We will add a more detailed analysis in the final version.

---

> > ### Author Rebuttal · Reviewer_atxW · 2026-04-03
> >
> > I thank the reviewers for their rebuttal. After reading their response I am happy to raise my rating.

---

> > > ### Author Response · Authors · 2026-04-04
> > >
> > > Dear Reviewer,
> > >
> > > Thank you very much for the positive feedback and **your decision to  raise the rating to accept**.
> > > We are writing to kindly bring to your attention a potential discrepancy in the system: while your comments indicate acceptance, the overall rating score appears unchanged(Weak reject). To ensure the editorial process proceeds smoothly before the deadline, would it be possible for you to check and adjust the score accordingly?
> > >
> > > We apologize for any inconvenience this may cause and thank you for your continued support.
> > >
> > > Best regards,
> > >
> > > Authors

---

### Official Review · Reviewer_4fxV · 2026-03-12

**Soundness:** 3
**Presentation:** 3
**Significance:** 3
**Originality:** 3
**Overall Recommendation:** 4
**Confidence:** 4

**Summary:**

The paper proposes a few-shot image classification approach that uses a Bayesian-inspired optimal integration framework (BOIF), treating pretrained models as priors. The paper considers the few-shot adaptation process of pretrained VLM as the estimation of the posterior conditioned on the downstream few-shot set. The authors hypothesize that the posterior logits can be obtained by summing the pre-trained model's output logits and the logits of a few-shot-based classifier.

**Compliance With Llm Reviewing Policy:**

Affirmed.

**Key Questions For Authors:**

Since LDC performs very close to the proposed approach in multiple cases, especially under the OOD setting, is there any advantage of the proposed approach over LDC, given that even the training time is similar.

**Limitations:**

Yes

**Strengths And Weaknesses:**

Soundness: Good

Presentation: Good

Significance: The problem addressed in the paper is significant. The proposed approach provides a new and interesting way of incorporating pretrained models for addressing the few-shot classification problem.

Originality: The contribution of the paper seems original.

Interesting and intuitive application of Bayesian posterior approximation to the few-shot learning task using a pretrained VLM.

Significant performance improvement over the compared methods.

Comprehensive ablation of the components of the approach has been carried out.

Since LDC performs very close to the proposed approach in multiple cases, especially under the OOD setting, is there any advantage of the proposed approach over LDC, given that even the training time is similar.

---

> ### Author Rebuttal · Authors · 2026-03-30
>
> **Q1**:  Since LDC performs very close to the proposed approach in multiple cases, especially under the OOD setting, is there any advantage of the proposed approach over LDC, given that even the training time is similar.
>
> **Reply**: Thank you for the insightful comment. While LDC achieves competitive performance in some settings, it is a heuristic design without theoretical justification. In contrast, our method is grounded in a principled Bayesian framework, which leads to a simpler and more interpretable model structure. Different from LDC, we use fixed weight $1/2$ to combine two pathways. Moreover, the designs of our modules $f_\theta(\cdot)$ and $g_\phi(\cdot)$ are different from LDC.
>
> Additionally, to further demonstrate the effectiveness of our approach, we evaluate it using a different backbone, DINOv2. Specifically, let $\mathbf{z}\_i= \text{Encoder}\_{DINOv2}(\mathbf{x}\_i) $ denote the extracted features. In Table R1, our method significantly outperforms the DINOv2 baseline, achieving more than $10\%$ improvement.
>
> At last, we perform more experiments in table R2, which further demonstrate the improvement of our method. These results provide additional evidence of the effectiveness of our approach. We will add this experiment to the revised paper.
>
>
> Table R1: Evaluation of the additive logit integration rule using DINOv2 as the backbone (10 datasets, 16-shot average accuracy).
>
> | Method | Average Accuracy (%) |
> | :--- | :--- |
> | DINOv2 backbone| 70.75 |
> | Ours | **81.79** |
>
> Table R2: Comparison of different methods under the OOD setting (ResNet-50 backbone). A higher value is better.
>
> | Method | Source (ImageNet) | Target (V2) | Target (Sketch) | Target (-A) | Target (-R) | Mean (Target) |
> | :--- | :--- | :--- | :--- | :--- | :--- | :--- |
> | LDC | 66.63 | 58.03 | 35.52 | 22.60 | 60.28 | 44.11 |
> | **Ours** | **66.84** | **58.45** | **36.00** | **22.93** | **60.50** | **44.47** |

---

> > ### Author Rebuttal · Reviewer_4fxV · 2026-04-04
> >
> > My concern regarding the advantage of the proposed approach over LDC is not resolved. The performance improvement even in the new setting is minimal (less than 0.5). So is there any distinct advantage of choosing the proposed method over LDC? Also considering reviewer UwgE's pending concerns, I will not change my rating currently.

---

### Official Review · Reviewer_Zakn · 2026-03-13

**Soundness:** 2
**Presentation:** 3
**Significance:** 3
**Originality:** 2
**Overall Recommendation:** 4
**Confidence:** 4

**Summary:**

The paper proposes a Bayesian-inspired Optimal Integration Framework (BOIF) for few-shot classification using pretrained Vision-Language Models (VLMs) like CLIP. The authors argue that current adaptation methods heuristically fuse pretrained knowledge and few-shot evidence. To provide a principled alternative, they formulate the pretrained model as a Bayesian prior and the few-shot support set as the likelihood. Under conditional independence approximations, they deduce that the optimal prediction is an additive combination of log-prior and log-likelihood. Guided by this, BOIF introduces a decoupled dual-path architecture: a prior pathway utilizing zero-shot CLIP logits augmented by a Multi-level Adapter Fusion (MAF) module, and a likelihood pathway based on a similarity-driven cache mechanism. The output logits are geometrically ensembled. Experiments across 11 benchmarks and Out-of-Distribution (OOD) settings show that BOIF achieves state-of-the-art accuracy compared to existing PEFT methods.

**Compliance With Llm Reviewing Policy:**

Affirmed.

**Final Justification:**

Given these merits, I believe the paper provides sufficient empirical value to the community. I will maintain my score of Weak Accept.

**Key Questions For Authors:**

yes

**Limitations:**

See weaknesses.

**Strengths And Weaknesses:**

Strengths:
1. The experimental validation is solid. Achieving 80.61% average accuracy at 16-shot across 11 diverse datasets, along with competitive OOD robustness on ImageNet-V2 and ImageNet-Sketch, demonstrates the practical effectiveness of the proposed architectural modifications.
2. Attempting to ground the heavily empirical field of prompt-tuning and adapter-fusion in a rigorous Bayesian framework is a highly commendable research direction.

Weaknesses:
1. The core claim of the paper is the explicit decoupling of the pretrained prior $f_\theta(x)$ and the support-set likelihood $g_\phi(x)$. However, the actual architectural design violates this completely. In Equation 12, the "prior" logit $\log f_\theta(x_i)$ is formulated as $o_i^{zs} + \beta W^T FC(z_i^e)$. The multi-level feature $z_i^e$ and the Feature Converter (FC) are explicitly optimized using the few-shot support set $D_{train}$. If the prior $f_\theta(x)$ is mathematically defined as $P(y|x, D_0)$ (conditioned only on pretraining data as per Eq. 8), injecting $D_{train}$-optimized features into it fundamentally breaks the Bayesian separation. The "prior" is no longer a prior; it is just another likelihood branch.Mathematical Looseness (Log-Probabilities vs. Logits): The theoretical derivation in Equation 10 correctly concludes that $\log P(y) \propto \log P_{prior} + \log P_{likelihood}$. However, the paper conflates unnormalized raw logits with log-probabilities. In Equation 11 and 12, $o_i^{zs}$ represents raw cosine similarity scaled by temperature (logits), but the authors directly denote it as $\log f_\theta(x_i)$. Furthermore, in Equation 15, the authors average the logits (multiplying by 1/2). Standard Bayesian updating requires the addition of log-prior and log-likelihood, not their arithmetic mean (which implies geometric ensembling). This disconnect renders the Bayesian derivation more of a post-hoc justification (math-washing) rather than a strict guiding principle.
2. Assumption 1 assumes that $D_0$ and $D_{train}$ are conditionally independent given $y$ and $x$. This is theoretically flawed in the context of foundation models. The input representation $x$ (specifically $z_i$) is directly extracted using the visual encoder trained on $D_0$. Thus, $x$ itself is heavily dependent on $\mathcal{D}_0$, making the conditional independence claim mathematically unsound.
3. Stripped of the Bayesian narrative, the actual method is effectively a combination of Tip-Adapter (cache module) and standard Multi-level Feature Fusion (MAF), ensembled via simple addition. This combination is highly incremental.

---

> ### Author Rebuttal · Authors · 2026-03-30
>
> **Q1**: The core claim of the paper is the explicit decoupling of the pretrained prior $f_\theta(x)$ and the support-set likelihood $g_\phi(x)$. However, the actual architectural design violates this completely. In Equation 12, the "prior" logit $\log f_\theta(x_i)$ is formulated as $o_i^{zs} + \beta W^T FC(z_i^e)$. The multi-level feature $z_i^e$ and the Feature Converter (FC) are explicitly optimized using the few-shot support set $D_{train}$. If the prior $f_\theta(x)$ is mathematically defined as $P(y|x, D_0)$ (conditioned only on pretraining data as per Eq. 8), injecting $D_{train}$-optimized features into it fundamentally breaks the Bayesian separation. The "prior" is no longer a prior; it is just another likelihood branch…….This disconnect renders the Bayesian derivation more of a post-hoc justification (math-washing) rather than a strict guiding principle.
>
> **Reply**: We thank the reviewer for the detailed technical comments. We would like to clarify the theoretical separation versus practical enhancement: Section 3.2 establishes that the pretrained prior and the support-set likelihood are independent components that should be combined via log‑probability addition under ideal conditions. In practice, however, the output of a pretrained model is only an approximation of the true Bayesian posterior, and the extracted features may not fully capture the information needed for downstream tasks. Therefore, we use the few-shot samples to fine-tune the pretrained model and enhance its representation quality, which is achieved by added these cross branches. We have already highlighted in lines 215—218(right column). We will clarify it better in the final version.
>
>
> **Q2**: Assumption 1 assumes that $D_0$ and $D_{train}$ are conditionally independent given $y$ and $x$. This is theoretically flawed in the context of foundation models. The input representation $x$ (specifically $z_i$) is directly extracted using the visual encoder trained on $D_0$. Thus, $x$ itself is heavily dependent on $\mathcal{D}\_0$, making the conditional independence claim mathematically unsound.
>
> **Reply**: For Assumption 1, in latent variable models [1], observed data are typically assumed to be generated from latent variables $z$, i.e., $P((x,y)|z)$. Consequently, both the training set and support set depend only on $z$, giving $P(\mathcal{D}\_0, \mathcal{D}\_{\text{train}}|z) = P(\mathcal{D}\_0|z)\cdot P(\mathcal{D}\_{\text{train}}|z)$. Under the common point estimation, we have $P(\mathcal{D}\_0, \mathcal{D}\_{\text{train}}|(x,y)) = P(\mathcal{D}\_0, \mathcal{D}\_{\text{train}}|z) P(z|(x,y))\propto P(\mathcal{D}\_0|(y,x))\cdot P(\mathcal{D}\_{\text{train}}|(y,x))$. Thus, the assumption holds even when the support set is drawn from the same visual world as the training set.
>
> Moreover, ref. [2] has shown that generalization ability stems from effective feature extraction rather than train-test similarity.
>
> We will discuss the reasonability of the assumption 1 in the final version.
>
> [1] DJ Bartholomew, et al., Latent variable models and factor analysis: A unified approach(book), 2011.
>
> [2] Mayilvahanan, Prasanna, et al., Does CLIP's generalization performance mainly stem from high train-test similarity?. ICLR, 2024.
>
>
> **Q3**: Stripped of the Bayesian narrative, the actual method is effectively a combination of Tip-Adapter (cache module) and standard Multi-level Feature Fusion (MAF), ensembled via simple addition. This combination is highly incremental.
>
> **Reply**:  We agree that our method seems like prior CLIP adaptation techniques. However, we would like to emphasize the difference of our method: prior methods typically require learning a tunable balancing parameter to combine the two pathways, which is challenging in few-shot settings due to limited number of samples. In contrast, our Bayesian derivation leads to a parameter-free equal-weight fusion rule, which is both principled and practically advantageous.
>
> Moreover, we conducted a sensitivity analysis by introducing a balancing factor $\rho$ into the integration equation: $ o_i^{out} = (1-\rho) \log f_\theta + \rho  \log g_\phi$. In Table R1, our method achieves the best performance at $\rho =0.5$. We will include this analysis in the revised paper.
>
> Table R1: Sensitivity analysis of the balancing coefficient $\rho$ (16-shot average accuracy over 11 datasets).
>
> | $\rho$| 0.30 | 0.35 | 0.40 | 0.45 | 0.50 | 0.55 | 0.60 | 0.65 | 0.70 |
> | :--- | :--- | :--- | :--- | :--- | :--- | :--- | :--- | :--- | :--- |
> | Accuracy (%) | 80.42 | 80.49 | 80.55 | **80.59** | **80.59** | 80.57 | 80.53 | 80.45 | 80.36 |

---

> > ### Author Rebuttal · Reviewer_Zakn · 2026-04-04
> >
> > The rebuttal confirms that the paper's theoretical framework is somewhat loosely coupled to its practical implementation (i.e., leaning towards post-hoc justification). Nevertheless, the empirical results are robust, the ablation studies are comprehensive, and the parameter-free fusion mechanism offers tangible practical benefits for few-shot adaptation.
> >
> > Given these merits, I believe the paper provides sufficient empirical value to the community. I will maintain my score of Weak Accept. I strongly suggest the authors incorporate the sensitivity analysis (Table R1) and the explicit distinction between "theoretical decoupling" and "practical enhancement" into the camera-ready version to improve the paper's theoretical honesty.

---

### Official Review · Reviewer_UwgE · 2026-03-18

**Soundness:** 3
**Presentation:** 3
**Significance:** 1
**Originality:** 2
**Overall Recommendation:** 4
**Confidence:** 4

**Summary:**

This paper proposes BOIF, a framework for few-shot classification that tries to give a Bayesian justification for combining pretrained and support-set signals. The argument goes: treat the pretrained model's output as a prior, treat the support set as providing a likelihood, assume conditional independence, and you get a simple rule—just add the two sets of logits. In practice the authors build this on top of CLIP with two parallel paths (one for the "prior," one for the "likelihood"), plug in a multi-level feature adapter and a lightweight cache module, and sum the logits at the end.

**Compliance With Llm Reviewing Policy:**

Affirmed.

**Final Justification:**

I thank the reviewers for their rebuttal. After reading their response I am happy to raise my rating.

**Key Questions For Authors:**

See the Weaknesses above.

**Limitations:**

The paper's own limitations discussion is relatively honest.

**Strengths And Weaknesses:**

Strengths

1. The most interesting part of the paper is not the final model, but the attempt to explain few-shot adaptation through a Bayesian lens.

2. Strong empirical package: The paper evaluates on 11 datasets, provides ablations, visualizations, and an efficiency table. This is a fairly complete empirical story for an adaptation paper.

3. Competitive performance: The reported averages over 11 datasets are strong.

4. The paper gives concrete training details, reports compute hardware, includes efficiency comparisons, and points to a released code repository. This improves reproducibility.


Weaknesses

1. The theoretical contribution is weaker than it appears: The paper's central selling point is the Bayesian derivation, but the result—additive logit combination—relies on four assumptions (conditional independence, weakly informative prior, pretrained model as approximate posterior, support-set module as likelihood estimator), each of which is strong and only approximately true. Critically, the conditional independence assumption (Assumption 1) is the load-bearing step, yet in reality the pretrained model and the support set are not independent information sources: the support set is drawn from the same visual world that the pretrained model has already seen, so their evidence about class identity is correlated. The paper acknowledges this but does not provide any empirical measurement of how far the actual data deviates from conditional independence, or how the method degrades as the assumption is increasingly violated. Without such analysis, the "principled" derivation is closer to a post-hoc justification of logit addition than to a genuinely predictive theory.

2. The practical instantiation is incremental over existing CLIP adaptation methods: Once the Bayesian framing is set aside, the concrete model is a CLIP backbone with a multi-level feature adapter and a simplified cache module—components that are directly related to Tip-Adapter, CLIP-Adapter, and multi-scale feature aggregation techniques that are standard in the literature. The final design principle—sum the logits of two pathways—is not far from what several existing methods already do implicitly (e.g., residual connections in CLIP-Adapter, cache-weighted additions in Tip-Adapter-F). The gap between what the theory promises (a new principle) and what the implementation delivers (a familiar two-stream CLIP adapter) is the paper's most significant weakness.

3. Generality is claimed but not demonstrated beyond CLIP: The paper frames BOIF as a general Bayesian integration principle applicable to any pretrained backbone, but every experiment uses CLIP. There is no evidence that the same additive-logit rule works with non-CLIP backbones such as DINOv2. This substantially weakens the paper's main claim of generality.

4. The paper does not adequately differentiate itself from concurrent and prior Bayesian few-shot work: Bayesian interpretations of few-shot learning are not new (e.g., Bayesian MAML, Bayesian prototypical networks). The paper does not position itself clearly against these earlier Bayesian perspectives, making it harder to assess what is genuinely novel beyond the specific CLIP instantiation.

5. Method innovation analysis: BOIF's main claim to novelty is the Bayesian derivation of the additive logit rule. However, logit-level combination of pretrained and adaptation pathways is already implicit in several existing CLIP adaptation methods—CLIP-Adapter uses residual addition of adapted features, Tip-Adapter-F adds cache-based logits to zero-shot logits, and even simple prompt-tuning methods effectively modify the prior logits while preserving the pretrained pathway. The Bayesian framing provides a cleaner narrative for why logit addition is reasonable, but it does not lead to a design that is structurally distinguishable from what practitioners already do. Compared with CLIP, CoOp, and Tip-Adapter, the novelty is best characterized as a theoretical reframing of an existing practice rather than a genuinely new adaptation mechanism. The multi-level feature adapter is a useful engineering addition, but multi-scale feature aggregation is well established and does not constitute a core methodological contribution.

---

> ### Author Rebuttal · Authors · 2026-03-30
>
> **Q1**： The theoretical contribution is weaker than it appears…four assumptions…
>
> **Reply**: Thank you for the analysis. We would like to emphasize that these assumptions are widely adopted and reasonable in practice:
>
> (1) For Assumption 1, in latent variable models [1], observed data are typically assumed to be generated from latent variables $z$, i.e., $P((x,y)|z)$. Consequently, both the training set and support set depend only on $z$, giving $P(\mathcal{D}\_0, \mathcal{D}\_{\text{train}}|z) = P(\mathcal{D}\_0|z)\cdot P(\mathcal{D}\_{\text{train}}|z)$. Under the common point estimation, we have $P(\mathcal{D}\_0, \mathcal{D}\_{\text{train}}|(x,y)) = P(\mathcal{D}\_0, \mathcal{D}\_{\text{train}}|z) P(z|(x,y))\propto P(\mathcal{D}\_0|(y,x))\cdot P(\mathcal{D}\_{\text{train}}|(y,x))$. Thus, the assumption holds even when the support set is drawn from the same visual world as the training set.
>
> (2) Assumption 2 corresponds to a random guess when no prior information about new classes is available, i.e., $P(y|\mathbf{x}) \approx \frac{1}{|\mathcal{Y}\_{\text{novel}}|} $, which is a standard and reasonable choice in practice.
>
> (3) For Assumptions 3 and 4, minimizing the cross-entropy loss is equivalent to maximizing the likelihood [2], making the outputs of neural models a natural approximation of the Bayesian posterior.
>
> In summary, all four assumptions hold in practices. We will add detailed discussions to clarify the validity in the final version.
>
> [1] DJ Bartholomew, et al., Latent variable models and factor analysis: A unified approach(book), 2011.
>
> [2] J Shore, On a relation between maximum likelihood classification and minimum relative-entropy classification, IEEE Transactions on Information Theory, 2003.
>
> **Q2**: The practical instantiation is incremental over existing CLIP adaptation methods…
>
> **Reply**: We agree that our CLIP adaptation method shares certain design similarities with existing approaches. However, our work provides the first theoretical justification of equally summing the logits from the two pathways, whereas prior methods typically require a tunable parameter to balance the contributions.
>
> In the few-shot setting, the limited number of samples makes it challenging to learn the parameter reliably. Our method solves this issue by the equal-weight fusion directly from Bayesian principles. To validate this point, we conduct a sensitivity analysis by introducing a balancing factor $\rho$: $o_i^{out} = (1-\rho) \log f_\theta + \rho  \log g_\phi$. In Table R1, the optimal performance is achieved at $\rho =0.5$, which aligns with our analysis. We will include this experiment in the final version.
>
>
> Table R1: Sensitivity analysis of $\rho$ (16-shot average accuracy over 11 datasets).
>
> | $\rho$| 0.30 | 0.35 | 0.40 | 0.45 | 0.50 | 0.55 | 0.60 | 0.65 | 0.70 |
> | :--- | :--- | :--- | :--- | :--- | :--- | :--- | :--- | :--- | :--- |
> | Accuracy (%) | 80.42 | 80.49 | 80.55 | **80.59** | **80.59** | 80.57 | 80.53 | 80.45 | 80.36 |
>
> **Q3**： Generality is claimed but not demonstrated beyond CLIP…such as DINOv2. …
>
> **Reply**: As suggested, we conducted additional experiments using DINOv2 as the backbone. In Table R2, our method significantly outperforms the DINOv2 baseline. We will add this experiment to the revised paper.
>
> Table R2: Performance of DINOv2 as the backbone (10 datasets, 16-shot average accuracy).
>
> | Method | Accuracy (%) |
> | :--- | :--- |
> | DINOv2 backbone| 70.75 |
> | Ours | **81.79** |
>
> **Q4**: The paper does not adequately differentiate itself from concurrent and prior Bayesian few-shot work…
>
> **Reply**: Our Bayesian-based method differs from prior work in two key aspects: (a) Methods such as Bayesian MAML [1] focus on learning a better initial parameters distribution for new tasks.
> (b) Approaches like Bayesian prototypical networks [2] use Bayesian networks as the pretrained neural models.
>
> In contrast, our method leverages Bayesian principles to derive an optimal combination of the two pathways. We will clarify this distinction in the revised paper.
>
>
> [1] Yoon, Jaesik, et al., Bayesian model-agnostic meta-learning. NeurIPS, 2018.
>
> [2] Gao CL, et al., BayProNet: Advancing Few-Shot Learning with Bayesian Prototype Network. CSAI, 2024.
>
>
> **Q5**: Method innovation analysis: BOIF's main claim to novelty is the Bayesian derivation of the additive logit rule….
>
> **Reply**: We would like to emphasize the key differences: prior methods typically require a tunable balancing parameter to combine the two pathways, which is challenging in few-shot settings due to limited data. In contrast, our Bayesian derivation leads to a parameter-free fusion rule, which is both principled and practically advantageous.
>
> To validate this, we conducted a sensitivity analysis by introducing a balancing factor $\rho$ in Table R1. The results verify the effectiveness of our method. We will include this analysis in the revised paper.

---

> > ### Author Rebuttal · Reviewer_UwgE · 2026-04-02
> >
> > I thank the authors for their effort. The new experiments are appreciated, but my two core concerns remain unresolved.
> >
> > First, the conditional independence argument (Q1) is circular—it assumes z fully mediates the dependence, which is precisely what needs empirical verification. I was hoping to see a correlation analysis between the two pathways' logits/errors, not a theoretical re-derivation.
> >
> > Second, the ρ-sensitivity analysis (Q2/Q5) actually weakens the paper's case: accuracy varies only ~0.2% across ρ∈[0.3, 0.7], meaning a practitioner could just pick ρ=0.5 as a default and lose nothing—no Bayesian theory needed. The flat curve undercuts the claim that the derivation provides a crucial practical advantage.
> >
> > On generality (Q3): the DINOv2 result compares against a vanilla backbone with no adapter. A fair comparison would be against Tip-Adapter-F or similar applied to DINOv2, so we can isolate what the Bayesian fusion rule actually contributes beyond standard adaptation.

---

> > > ### Author Response · Authors · 2026-04-03
> > >
> > > Thank you for the detailed response. We have carried out more experiments to address the comments. We hope that our experiments could address you concerns.
> > >
> > > **Q6**：First,  the conditional independence argument (Q1) is circular—it assumes z fully mediates the dependence, which is precisely what needs empirical verification. I was hoping to see a correlation analysis between the two pathways' logits/errors, not a theoretical re-derivation.
> > >
> > > **Reply**: We have performed a correlation analysis as suggested. Specifically, we computed the Pearson correlation between the logits of the two pathways across all samples. The overall average correlation is **0.34**. Moreover, the correlation distribution is shown in Table R3. The results indicate weak correlation of the two pathways’ logits, providing empirical support for our conditional independence assumption. We will add the correlation results to the final version, as suggested.
> > >
> > > Table R3: The Pearson correlation between the two pathways' logits for all samples from all datasets.
> > > | Range | 0-0.1 | 0.1-0.2 | 0.2-0.3 | 0.3-0.4 | 0.4-0.5 | 0.5-0.6 | 0.6-0.7 | 0.7-0.8 | 0.8-0.9 | 0.9-1.0 |
> > > |----------|-------|---------|---------|---------|---------|---------|---------|---------|---------|----------|
> > > |Fraction of samples      | 4.5%  | 13.6%   | **23.4%**   | 22.1%   | 15.3%   | 6.0%    | 1.5%    | 1.6%    | 3.6%    | 3.3%     |
> > >
> > >
> > >
> > > **Q7**: Second, the ρ-sensitivity analysis (Q2/Q5) actually weakens the paper's case: accuracy varies only ~0.2% across ρ∈[0.3, 0.7], meaning a practitioner could just pick ρ=0.5 as a default and lose nothing—no Bayesian theory needed. The flat curve undercuts the claim that the derivation provides a crucial practical advantage.
> > >
> > > **Reply**: The sensitivity to $\rho$ depends on both the number of $K$($K$-shot) and the dataset. Table R2 shows results for 16-shot. For smaller $K$, the variation becomes much more pronounced. For example, Table R4 presents the sensitivity for 4-shot on UCF101, where the optimal accuracy is achieved at $\rho=0.5$, but performance degrades noticeably away from this value at $\rho=0.55$. Moreover, sensitivity can vary substantially across different $K$-shot settings. Therefore, our method provides a principled solution. We will include the relevant sensitivity experiments and discussion in the final version.
> > >
> > > Table R4: Sensitivity analysis of $\rho$ for $4$-shot in dataset UCF101.
> > >
> > > | ρ | 0.30 | 0.35 | 0.40 | 0.45 | **0.50** | 0.55 | 0.60 | 0.65 | 0.70 |
> > > |---|---|---|---|---|---|---|---|---|---|
> > > | Acc(%) | 75.79 | 76.05 | 76.05 | 76.10 | **76.21** | 75.89 | 75.76 | 75.18 | 74.73 |
> > >
> > >
> > > **Q8**: On generality (Q3): the DINOv2 result compares against a vanilla backbone with no adapter. A fair comparison would be against Tip-Adapter-F or similar applied to DINOv2, so we can isolate what the Bayesian fusion rule actually contributes beyond standard adaptation.
> > >
> > > **Reply**: We have compared our method against both DINOv2 and Tip-Adapter-F in Table R5. In Table R5, our method achieves 81.79\% average accuracy, outperforming both adapter-based baselines. We will add these results to the final version, as suggested.
> > >
> > > Table R5: Performance comparison using DINOv2 as backbone (10 datasets, 16-shot average accuracy).
> > >
> > > | Method | Accuracy (%) |
> > > |--------|--------------|
> > > | DINOv2+prototype | 70.75 |
> > > | Tip-Adapter-F | 81.60 |
> > > | Ours | **81.79** |

---

### Decision · Program_Chairs · 2026-04-30

**Decision:**

Accept (regular)

**Comment:**

The submission proposes a new approach for few-shot learning inspired by Bayesian inference. Several assumptions, when combined with Bayes' rule, are used to show that one can decompose the problem of few-shot classification into adding the logits of a pre-trained "prior" model and a likelihood model. The submission provides a very extensive experimental comparison across a number of datasets and compares to quite a few related methods.

The reviewers identified a number of issues in their original reviews. These included questions about the validity of the assumptions, the incremental improvement in performance over previous methods, the restriction of the evaluation to CLIP, and that it does not do a good job comparing and contrasting to other work in the extensive literature on Bayesian few-shot learning. The authors provided substantial new results during the rebuttal phase, and clarified some of the claims made in the paper. The reviewers found these arguments quite convincing, and the paper now has generally positive assessments.